# The Occurrence of Quill Mites (Arachnida: Acariformes: Syringophilidae) on Bee-Eaters (Aves: Coraciiformes: Meropidae: *Merops*) of Two Sister Clades

**DOI:** 10.3390/ani11123500

**Published:** 2021-12-08

**Authors:** Maciej Skoracki, Jakub Z. Kosicki, Bozena Sikora, Till Töpfer, Jan Hušek, Markus Unsöld, Martin Hromada

**Affiliations:** 1Department of Animal Morphology, Faculty of Biology, Adam Mickiewicz University, Uniwersytetu Poznańskiego 6, 61-614 Poznań, Poland; boszka@amu.edu.pl; 2Department of Avian Biology and Ecology, Faculty of Biology, Adam Mickiewicz University, Uniwersytetu Poznańskiego 6, 61-614 Poznań, Poland; kubako@amu.edu.pl; 3Section Ornithology, Zoological Research Museum Alexander Koenig, Leibniz Institute for the Analysis of Biodiversity Change, Adenauerallee 127, 53113 Bonn, Germany; t.toepfer@leibniz-zfmk.de; 4National Museum, Václavské náměstí 68, 11579 Praha 1, Czech Republic; jan.husek@nm.cz; 5Zoologische Staatssammlung München, Sektion Ornithologie, Münchhausenstr. 21, 81247 München, Germany; unsoeld@snsb.de; 6Laboratory and Museum of Evolutionary Ecology, Department of Ecology, Faculty of Humanities and Natural Sciences, University of Presov, 08001 Prešov, Slovakia; 7Faculty of Biological Sciences, University of Zielona Góra, 65-516 Zielona Góra, Poland

**Keywords:** Acari, birds, ectoparasites, host-shift, zoogeography

## Abstract

**Simple Summary:**

Parasitic mites of the family Syringophilidae (quill mites) represent the most diverse prostigmatan family associated with birds. Here, we aim (i) to investigate quill mites of a well-defined monophyletic clade of nine bee-eater species, containing mostly migratory African and Asian species; (ii) to establish ectoparasite geographic ranges; and (iii) to discuss patterns of host-parasite relationships and possible host-switches. We have found that despite quill mites being highly specific on the host species or genus level, host-switches may occur, particularly when the ecology of host species overlaps.

**Abstract:**

We studied the quill mite fauna of the family Syringophilidae, associated with bee-eaters. We examined 273 bird specimens belonging to nine closely related species of the genus *Merops*, representing two phylogenetic sister clades of a monophyletic group. Our examination reveals the presence of two species of the genus *Peristerophila*, as follows: (1) a new species *Peristerophila mayri* sp. n. from *Merops viridis* in the Philippines, *M. leschenaulti* in Nepal and Sri Lanka, and *M. orientalis* in Sri Lanka; and (2) *P. meropis* from *M. superciliosus* in Tanzania and Egypt, *M. persicus* in Sudan, Tanzania, Liberia, Senegal, Kenya, and D.R. Congo, *M. ornatus* in Papua New Guinea, *M. philippinus* in Thailand, Indonesia and Sri Lanka, and *M. americanus* in the Philippines. The prevalence of host infestations by syringophilid mites varied from 3.1 to 38.2%. The distribution of syringophilid mites corresponds with the sister clade phylogenetic relationships of the hosts, except for *P. meropis* associated with *Merops americanus*. Possible hypotheses for the host lineage shift are proposed.

## 1. Introduction

The family Syringophilidae (Arachnida: Acariformes: Prostigmata: Cheyletoidea) represents the most diverse prostigmatan family associated with birds. To date, there are about 400 described species, and the range of their hosts comprises 24 orders and 95 families from all zoogeographical regions, except Antarctica [1]. All species of this family are permanent and obligatory ectoparasites of birds, and most of them display a high degree of host specificity, being mono- or oligoxenous parasites [2,3,4,5].

The bee-eaters (Aves: Coraciiformes: Meropidae) comprise a group of 3 genera and up to 28 species (depending on taxonomy), distributed throughout the palaeotropics and southern Eurasia, and Oriental and Australian region (itself, without Oceania). They are found in habitats ranging from dry savannas to rainforests [6,7,8]. Despite the highest recent diversity of bee-eaters in Africa (22 species), the origin of this family possibly lies in Asia from where bee-eaters dispersed to Africa, then to other regions, and even back to Asia [9,10].

Previous records of syringophilid mites associated with bee-eaters include four species: two species of the genus *Syringophilopsis* Kethley, 1970 (*S. albicollisi* Skoracki and Dabert, 2000 and *S. melittophagi* Skoracki and Dabert, 2001), and two species of the genus *Peristerophila* Kethley, 1970 (*P. meropis* Skoracki et al. 2017 and *P. hirundinis* Skoracki et al. 2020) [11,12,13,14].

Here, we aim to (i) investigate quill mites of a well-defined monophyletic clade of nine bee-eater species containing mostly migratory African and Asian species [9]; (ii) establish ectoparasite geographic ranges; and (iii) discuss patterns of host-parasite relationships and possible host switches. This paper is the continuation of the studies on syringophilid mites parasitizing poorly explored birds of the family Meropidae [13,14].

## 2. Materials and Methods

The mite material used in this study was collected according to the technique proposed by Skoracki (2011) [3] from quills of under-tail coverts of dry bird skins housed in the following ornithological collections: Bavarian State Collection of Zoology (Munich, Germany) (ZSM), Royal Museum for Central Africa (Tervuren, Belgium) (RMCA), Zoological Research Museum Alexander Koenig (Bonn, Germany) (ZFMK), National Museum, Nairobi, Kenya (NMK), and National Museum, Prague, Czechia (NMP).

We investigated all nine species of the monophyletic host clade, as identified by Marks et al. [9] (Figure 1). A total of 273 individuals of *Merops americanus* (*n* = 8), *M. apiaster* (*n* = 75), *M. leschenaulti* (*n* = 13), *M. orientalis* (*n* = 32), *M. ornatus* (*n* = 25), *M. persicus* (*n* = 68), *M. philippinus* (*n* = 8), *M. superciliosus* (*n* = 34), and *M. viridis* (*n* = 10) were examined for the presence of quill mites of the family Syringophilidae. From each bird specimen, we removed about 5–7 under-tail coverts which are commonly occupied by *Peristerophila* mites on the *Merops* hosts [13]. All collected feathers were checked under the stereomicroscope and opened using fine forceps. Infested feathers were placed in Eppendorf’s tubes with Nesbitt’s solution at 40 °C for about 24 h, and then mites were mounted on microscope slides in Hoyer’s medium. Slide-mounted mites were examined under a light microscope (ZEISS Axioscope2™; Carl Zeiss AG, Jena, Germany) equipped with DIC optics and camera lucida.

In the descriptions, the idiosomal setation follows Grandjean [15], as adapted for Prostigmata by Kethley [16]. The nomenclature of leg chaetotaxy follows that proposed by Grandjean [17]. Measurements (ranges) for paratypes are given in parentheses following data for the holotype. All measurements are given in micrometers. Depositories of mite specimens are given using the following abbreviations: AMU—Adam Mickiewicz University, Department of Animal Morphology, Poznan, Poland; ZSM—Bavarian State Collection of Zoology, Munich, Germany.

To visualize host phylogeny, a tree was constructed based on data available from http://birdtree.org/ (accessed on 28 October 2021) [18], using the “Ericson All Species tree” with 1000 randomly generated trees. Currently, this tool is widely used in bird evolutionary ecology studies (e.g., [19,20]), including the investigation of the host phylogenies of bird parasites [21,22]. The most credible tree was then determined using TreeAnnotator v1.8.2 in the software BEAST v1.8.2 [23]. The consensus tree was then graphically adjusted in FigTree v1.4.2 [24].

Host data ranges were taken from BirdLife International [25]. The map was drawn using QGIS 3.16 software (Open Source Geospatial Foundation, Beaverton, Oregon, United States) with WGS 84 geographical projection.

## 3. Results

### 3.1. Species Composition

In total, we investigated 273 host individuals belonging to nine *Merops* species, of which 56 specimens (21.1%) were infested by quill mites, whose prevalence varied from 3.1% to 38.2% (Table 1).

We detected two quill mite species of the genus *Peristerophila*: *P. mayri* sp. n. and *P. meropis* (Skoracki et al., 2017). Their occurrence on the respective hosts corresponds with phylogenetic relationships within the genus *Merops*, as identified by Marks et al. [9]: the subclade including *M. ornatus*, *M. apiaster*, *M. persicus*, *M. superciliosus*, and *M. philippinus* is infested by *P. meropis,* and the subclade including *M. viridis*, *M. leschenaulti*, and *M. orientalis* is infested by *P. mayri*. Unexpectedly, *P. meropis* is found on *M. americanus* belonging to the second subclade (Figure 1).

We describe a new mite species, *Peristerophila mayri* sp. n., infesting quill feathers of three bee-eater species: *Merops viridis*, *M. leschenaulti*, and *M. orientalis*. The other five analyzed species, i.e., *M. americanus*, *M. ornatus*, *M. persicus*, *M. philippinus*, and *M. superciliosus*, represent new host species for the recently described *Peristerophila meropis* (Table 1).

### 3.2. Systematics

Family—Syringophilidae Lavoipierre, 1953.

Subfamily—Syringophilinae Lavoipierre, 1953.

Genus—*Peristerophila* Kethley, 1970.

#### 3.2.1. Peristerophila mayri Skoracki and Hromada sp. n.

##### Female, Heteromorphic Form, Holotype

Description. Total body length 1120 (1100–1150 in four paratypes). *Gnathosoma*. Stylophore apunctate, covered with longitudinally striate ornament, 155 (140–155) long. Movable cheliceral digit 100 (100–105). Each medial branch of peritremes with two or three chambers, each lateral branch with four chambers. Infracapitulum apunctate. *Idiosoma*. Propodonotal shield weakly sclerotised, apunctate, divided into two lateral sclerites bearing bases of setae *ve* and *si* and unpaired narrow medial sclerite. Setae *ve* and *si* subequal in length. Bases of setae *c1* situated posterior to level of setal bases *se*, setae *c2* situated anterior to level of setal bases *se*. Hysteronotal shield absent. Setae *d1*, *d2*, and *e2* subequal in length. Pygidial shield well developed, apunctate, with rounded anterior margin. Genital (*g1–g2*) and pseudanal (*ps1–ps2*) setae subequal in length. Genital plate absent. Length ratio of setae *ag1*:ag2:ag3 4.6:1:4.6–5.2. Coxal fields I punctate, II–IV sparsely punctate. Setae *3c* 3–4 times longer than 3b. *Legs*. Fan-like setae *p′* and *p″* of legs III–IV with 11–13 tines. Tarsi I densely punctate, II–IV sparsely punctate. Setae *tc″III–IV* about twice as long as *tc'III–IV*. Apodemes I 110 (110–120) long, apodemes II 50 (45–50) long. *Lengths of setae*: *ve* 20 (20–25), *si* 25 (25–30), *se* 260 (255), *c1* (220–225), *c2* 235 (220–245), *d1* 235 (220–235), *d2* 255 (230–255), *e2* 245 (235–255), *f1* 20 (20), *f2* (280–290), *h1* 20 (20–25), *h2* (330), *ag1* 115 (110–125), *ag2* 25 (25), *ag3* 130 (125), *3b* 35 (35–40), *3c* 135 (105–125), *tc′III–IV* 35 (25–35), *tc"III–IV* 75 (70–75), *ps1* and *ps2* 15 (12–15), *g1* and *g2* 12 (12–15), *l′RIII* 40 (35–40), and *l′RIV* 30 (25–30) (Figure 2 and Figure 3).

##### Homeomorphic Female and Male

Not found.

##### Type Material 

Female holotypes and four female paratypes from Blue-throated Bee-eater *Merops viridis* Linnaeus (Meropidae); Philippines: Luzon Isl., 30 August 1890 (host specimen in the ZSM, uncatalogued).

##### Type Material Deposition 

All type specimens are deposited in the AMU (reg. no. AMU-SYR.608), except one female paratype, which is deposited in the ZSM (reg. no. ZSMA-20190423).

##### Additional Material

Ex Chestnut-headed Bee-eater *Merops leschenaulti* (host in the ZSM, uncatalogued), Nepal: Hitara, 22 March 1962, coll. W. Lagew; two females (heteromorphic form – HET) deposited in the AMU (reg. no AMU-SYR. 598) and one female (HET) in the ZSM (reg. no. ZSMA20190424). Ex the same host species (host in the ZSM, uncatalogued); Sri Lanka: Anuradhapura, 6 January 1905, coll. Doflein; three females (HET) deposited in the AMU (reg. no AMU-SYR. 597) and one female (HET) in the ZSM (ZSMA20190424).

Ex Little Green Bee-eater *Merops orientalis* (host in the ZSM, uncatalogued); Sri Lanka: Anuradhapura, 7 January 1905, coll. Doflein; two females (HET) deposited in the AMU (reg. no. AMU-SYR.578).

##### Habitat

Quills of under- and upper-tail covers and contour feathers of cloaca region.

##### Etymology 

This species is named in honor of Ernst Walter Mayr (1904–2005), arguably the most preeminent evolutionary biologist of the twentieth century. Mayr’s contributions during the course of his remarkable life are manifold; among these, he had a lifelong special interest in ornithology and biogeography (e.g., [26]). When investigating the evolution of populations and species, he also referred to bee-eaters studied in this paper in his works (e.g., [27]).

##### Differential Diagnosis 

*Peristerophila mayri* sp. n. is morphologically most similar to *P. coraciidus* Skoracki, Hromada et Sikora, 2020, described from Dollarbird *Eurystomus orientalis* (Coraciidae) from Papua New Guinea [28]. In females of both species, the hysteronotal shield is absent, and each medial branch of the peritremes has two or three chambers, whereas each lateral branch has four chambers. This new species differs from *P. coraciidus* by the following features: in heteromorphic females of *P. mayri*, the medial propodonotal shield is present; the tarsal fan-like setae of legs III and IV have 11–13 tines; genital (*g1–g2*) and pseudanal (*ps1–ps2*) setae are subequal in the length; the lengths of setae *ag1*, *ag2*, and *ag3* are 110–125, 25, and 125–130, respectively; the pygidial shield is apunctate, well developed and with the rounded anterior margin. In heteromorphic females of *P. coraciidus*, the medial propodonotal shield is absent; the tarsal fan-like setae of legs III and IV have 17–19 tines; genital setae (*g1–g2*) are 4–5 times longer than pseudanal setae (*ps1–ps2*); the lengths of setae *ag1*, *ag2*, and *ag3* are 180–265, 65–85, and 255–265, respectively; the pygidial shield is punctate and reduced to small region bearing bases of setae *f1* and *f2*, with indiscernible anterior margin.

#### 3.2.2. Peristerophila meropis (Skoracki, Hromada et Sikora, 2017)

*Castosyringophilus meropis* Skoracki et al., 2017: 2, Figs 1 and 2.

*Peristerophila meropis*, Skoracki et al. 2020: 1816 [new combination].

Type host: *Merops apiaster* (Meropidae).

##### Hosts and Distribution 

European Bee-eater *Merops apiaster* from Spain, France, Gibraltar, Italy, Macedonia, Romania, Bosnia and Herzegovina, Greece, Turkey, Russia, Azerbaijan, Pakistan, Morocco, Tanzania, and Kenya [13]; Rufous-crowned Bee-eater *Merops americanus* (new host) from the Philippines, Rainbow Bee-eater *Merops ornatus* Latham (new host) from Papua New Guinea, Olive Bee-eater *Merops superciliosus* (new host) from Egypt and Tanzania; Blue-cheeked Bee-eater *Merops persicus* (new host) from Sudan, Tanzania, Liberia, Senegal, Kenya, and D.R. Congo, and Blue-tailed Bee-eater *Merops philippinus* from Thailand, Indonesia, and Sri Lanka (current study).

##### Habitat

Quills of under- and upper-tail covers, lesser wing coverts, back contour feathers, and contour feathers of cloaca region.

##### New Material Examined

There were 2 females (heteromorphic form – HET) and 1 male from *Merops superciliosus* (host in the ZSM; reg. no. 64.909); Tanzania: Pwani Region, Kibaha District, Soga, 10 September 1960, coll. Th. Andersen; all mite specimens deposited in the AMU (reg. no. AMU-SYR.599), except 1 female (HET) which is deposited in the ZSM (reg. no. ZSMA20190418). There were 3 females (HET), 2 males, 7 tritonymphs, 6 protonymphs, and 2 larvae from the same host species (host in the ZSM; uncatalogued); Egypt: near Lake Qarun, 20 March 1989, coll. U. Norra; all mite specimens deposited in the AMU (reg. no. AMU-SYR. 599B), except 1 female (HET), 1 tritonymph, and 2 protonymphs in the ZSM (reg. no. ZSMA20190419).

There were 7 females (HET), 1 female (homeomorphic form – HOM), 1 male, 3 tritonymphs, 3 protonymphs from *Merops persicus* (host in the ZSM; reg. no. 12.2275, male); Sudan: area of White Nil, 25 February 1912, coll. Hesselberger; all mite specimens deposited in the AMU (reg. no. AMU-SYR.845), except 2 females (HET) in the ZSM (reg. no. ZSMA20190420). There were 2 females (HET) from the same host species (host in the ZSM; reg. no. 64.906, male); Tanzania: Pwani Region, Kibaha District, Soga, 30 January 1962, coll. Th. Andersen; all mite specimens deposited in the AMU (reg. no. AMU-SYR. 740B). There was 1 female (HET), 3 protonymphs from the same host species (host in the ZSM; reg. no. 64.907, female); Tanzania: Dodoma District, Bahi, alt. 3000ft, 6 March 1953, coll. Th. Andersen; all mite specimens deposited in the AMU (reg. no. AMU-SYR.740). There was 1 female (HET) from the same host species (host in the ZSM; reg. no. 08.661); Liberia: “Pessyland”, 1909, coll. J. Scherer; mite specimen deposited in the AMU (reg. no. AMU-SYR.749). There were 2 females (HET) from the same host species (host in the ZSM; reg. no. A659); Senegal: 1837, no other data; all mite specimens deposited in the AMU (reg. no. AMU-SYR.741). There was 1 female from the same host species (host in the RMCA; reg. no. 97829/D147, male); Kenya: 7 January 1918, coll: van Somevn; specimen deposited in the AMU (reg. no. AMU-SYR.743). There were 4 females (HET), 4 females (HOM), 1 male, 8 tritonymphs, 4 protonymphs, 5 larvae from the same host species (host in the RMCA; reg. no. 43985/D1); D.R. Congo: Katanga Prov., Kadima, February 1948, coll. Dewit; all mite specimens deposited in the AMU (reg. no. AMU-SYR.742). There were 2 females (HET) from the same host species (host in the RMCA; reg. no. 7964/D135); D.R. Congo: Bas Congo Prov., 1915, coll. van Saughem; all mite specimens deposited in the AMU (reg. no. AMU-SYR.748). There were 3 females (HET), 1 larva from the same host species (host in the RMCA; reg. no. 9766/D135); D.R. Congo: Bas Congo Prov., 15 February 1922, coll. Schahnden; all mite specimens deposited in the AMU (reg. no. AMU-SYR.747). There were 1 female (HET), 2 males from the same host species (host in the RMCA; reg. no. 36649/D146); D.R. Congo: Orientale Prov., Ituri, 7 November 1939, coll. Lepersonn; all mite specimens deposited in the AMU (reg. no. AMU-SYR.744).

There were 2 females (HET), 3 larvae from *Merops philippinus* (host in the NMP; reg. no. P6V 102636); Thailand: Phuket, no other data; all mite specimens deposited in the AMU (reg. no. AMU-SYR.1001). There were 5 females (HET) from the same host species (host in the ZSM; reg. no. A661/V341); Sri Lanka: Maddawadi, 9 January 1905, coll. F. Doflein; all mite specimens deposited in the AMU (reg. no. AMU-SYR.1002). There were 4 females (HET) from the same host species (host in the ZSM; reg. no. 25.94/V343); Indonesia: Java, no other data; all mite specimens deposited in the AMU (reg. no. AMU-SYR.1003).

There were 6 females (HET), 2 tritonymphs, 4 protonymphs, 9 larvae, from *Merops ornatus* (host in the ZSM; reg. no. 13.97/V325); Papua New Guinea: Morobe Prov., near Finschhafen, coll. Hahl; all mite specimens deposited in the AMU (reg. no. AMU-SYR.877), except 1 female (HET) in the ZSM (reg. no. ZSMA20190421). There were 2 females (HET), 1 male, 1 tritonymph, 1 protonymph from the same host species (host in the ZSM; reg. no. 10.120/V326); Papua New Guinea: Madang Prov., Stephansort, coll. M. Hofwokel; all mite specimens deposited in the AMU (reg. no. AMU-SYR.877B). There were 1 female (HET), 1 male, 1 tritonymph, 1 protonymph, 1 larva from the same host species (host in the ZSM; reg. no. 11.671/V323); Papua New Guinea: 17 March 1910, coll. L. Wiedenfeld; all mite specimens deposited in the AMU (reg. no. AMU-SYR.877C). There was 1 female (HET) from same host species (host in the ZSM; reg. no. 11.676/V324); Papua New Guinea: August 1910, coll. L. Wiedenfeld; mite specimen deposited in the AMU (reg. no. AMU-SYR.877D).

There were 12 females (HET) and 2 tritonymphs from *Merops americanus* (host in the ZFMK; reg. no. ZFMK_ORN_1965.814/B3); Philippines: Luzon Isl., Abra Prov., May 1965, coll. unknown; all mite specimens deposited in the AMU (reg. no. AMU-SYR.594). There were 3 females (HOM) and 3 tritonymphs from the same host species (host in the ZFMK; reg. no. ZFMK_ORN 1965.813/B2) and locality, April 1965, coll. unknown; all mite specimens deposited in the AMU (reg. no. AMU-SYR.594B). There were 1 female (HET), 1 male, 5 tritonymphs, 4 protonymphs, 3 larvae from same host species (host in the ZFMK; reg. no. ZFMK_ORN 1966.1209/B1); Philippines: Mindoro Isl., July 1964, coll. Bregulla; all mite specimens deposited in the AMU (reg. no. AMU-SYR.594C).

## 4. Discussion

Syringophilid mites are tightly associated with their hosts, where most of them represent mono- or oligoxenous species limited to a particular species or group of phylogenetically closely related hosts [3,4]. Because the biology of these obligate and permanent parasites is heavily linked to their avian hosts, their diversification pattern often reflects those of their hosts according to the old, but still actual rules of Fahrenholz [29] and Eichler [30].

In our study, birds belonging to two sister clades of bee-eaters are infested by two species of syringophilid mites (Figure 1 and Figure 4), which shows that the occurrence of *Peristerophila* mites indeed tightly corresponds with the phylogenies of their avian hosts, except for the presence of *Peristerophila meropis* on *Merops americanus*.

Until recently, *M. americanus* was treated as a subspecies of the Blue-throated Bee-eater *M. viridis*, i.e., *M. viridis americanus* [9,31,32,33,34]. However, recently, this species was elevated to a full species, the Rufous-crowned Bee-eater [7,8,35,36,37]. The geographic range of *M. viridis* stretches from S China, Thailand, and Indochina to Sumatra, Borneo, and Java, whereas *M. americanus* is confined to the Philippines, thus, they are vicariants [7,36].

We are describing an interesting situation, when two closely related host species, considered until recently to be subspecies of one species, are parasitized by different ectoparasitic quill mites. We suggest the following hypothesis why *M. americanus* is parasitized by *P. meropis*, a quill mite species typical for the sister bee-eater subclade: *M. americanus* occurs sympatrically with *M. philippinus* on the Philippines archipelago (Figure 4), where they often dwell in the same habitats. Since *M. philippinus* is a highly colonial species with tens or hundreds of pairs breeding together, they sometimes form mixed colonies with other local bee-eater species. Both *M. philippinus* and *M. americanus* breed in ground nest-burrows [38,39] that might be re-used over the years; it is also probable that different species use the same hole. Thus, the infestation of *M. americanus* by *P. meropis* could be explained by a host-switch between the two sympatric bee-eater species via their colonial nesting sites.

Due to the host phylogenetic affinities, it seems obvious to expect that *M. americanus* should be parasitized by *P. mayri*; however, it is not possible to speculate at the moment, whether *P. mayri*, maybe once infesting *M. americanus* and/or its common ancestor with *M. viridis*, was outcompeted when occurring sympatrically with *M. meropis* in the same habitat and host (Figure 1 and Figure 4). More relevant data on the biology and ecology of both ectoparasites and their hosts are needed.

Interestingly, there is another, even more distantly related bird species parasitized by *Peristerophila meropis* living sympatrically on the Philippines—the Collared Kingfisher *Todiramphus chloris* (Coraciiformes: Alcedinidae) [26]—which, besides nest-holes excavated in arboreal termitaria, rotten tree trunks, etc., also breeds in earth banks and in the ground, sometimes in nest-burrows excavated by another bird species [40]. Until now, it is the only known non-meropid host of this widespread quill mite taxon. However, for the time being, it is not possible to conclude if this kingfisher is another example of an isolated host-switching event, or *P. meropis* can be established also on other kingfishers, without detailed investigations of quill mites on other related hosts.

Mapping the ranges of both investigated quill mite species enabled us also to denote their zoogeographic demarcations. Easternmost boundaries of *P. mayri* range are within the classical Wallace line (after Huxley), delineated with the extent of land at the time of the last glacial maximum. Hosts of *P. mayri* are residents [6,8,9]. On the other hand, the range of *P. meropis* is crossing these natural boundaries as far as to Australia; all hosts of *M. meropis* but *M. americanus* (which is the only species from different clade) are migratory, some of them long-distance migrants [6,8,9].

Another interesting fact in our parasite-host study system is not only the relatively high prevalence of infestation, compared to other wild avian taxa [21,41,42,43,44,45,46,47], but a situation when actually all taxa in the whole host clade are parasitized by quill mites. We never recorded a similar situation during our studies of quill mites, it is rather usual that some hosts, despite being sufficiently sampled, are not parasitized [21,42,43,44]. We expect that such high prevalence and presence on all host species are related to bee-eaters high sociality, when up to hundreds of pairs are breeding in the same colony; cooperative breeding, when, besides parents, several helpers are participating in the raising of the offspring in the nest-burrow; and finally, bee-eaters tendency to perch often touching each other [6,8,38,39].

## Figures and Tables

**Figure 1 animals-11-03500-f001:**
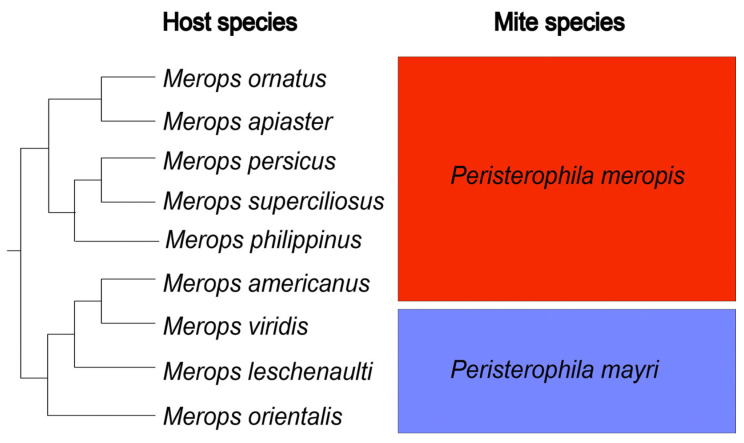
Phylogenetic reconstruction of the bee-eater lineage under study (according to Marks et al. [9]) and associated quill mite species.

**Figure 2 animals-11-03500-f002:**
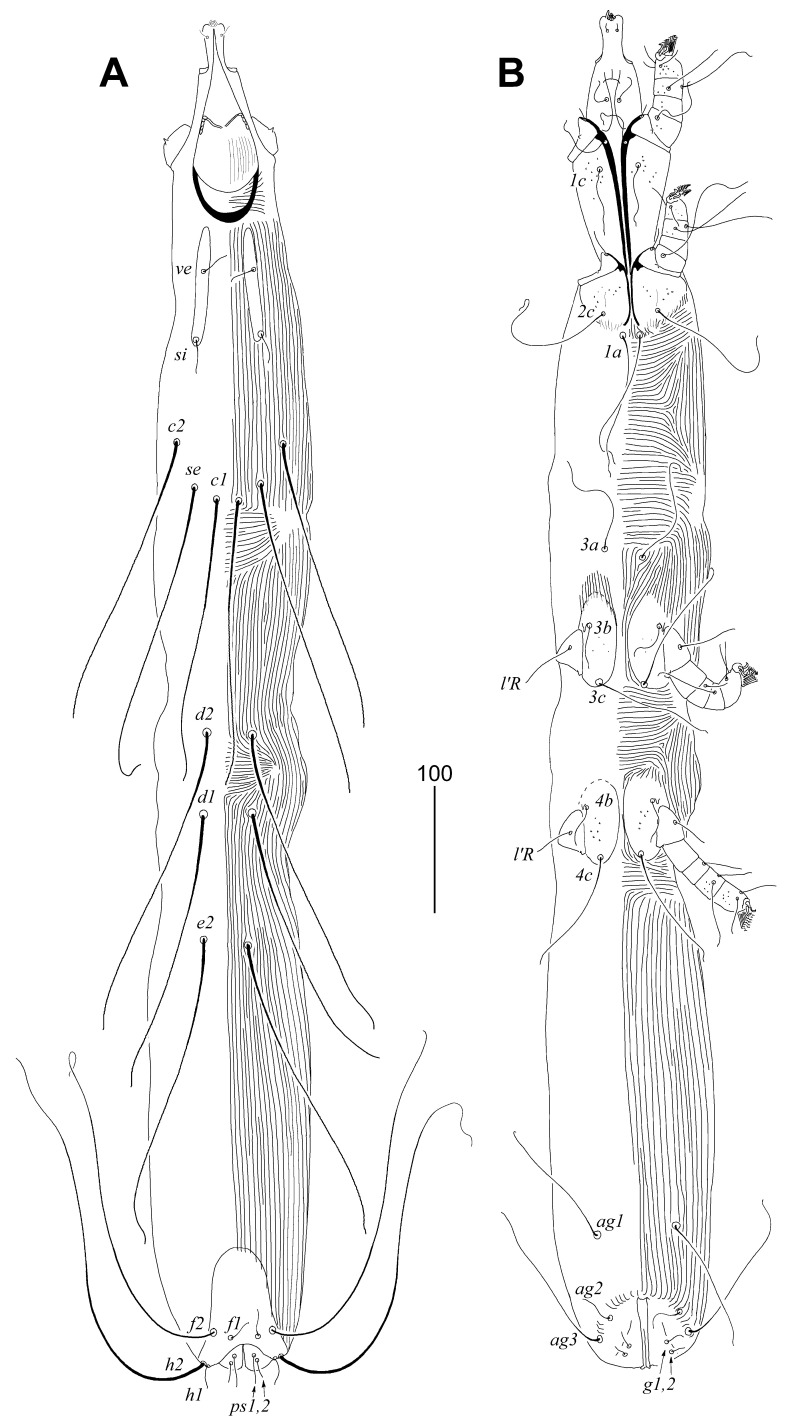
*Peristerophila mayri* Skoracki and Hromada sp. n., female. (**A**)—dorsal view; (**B**)—ventral view.

**Figure 3 animals-11-03500-f003:**
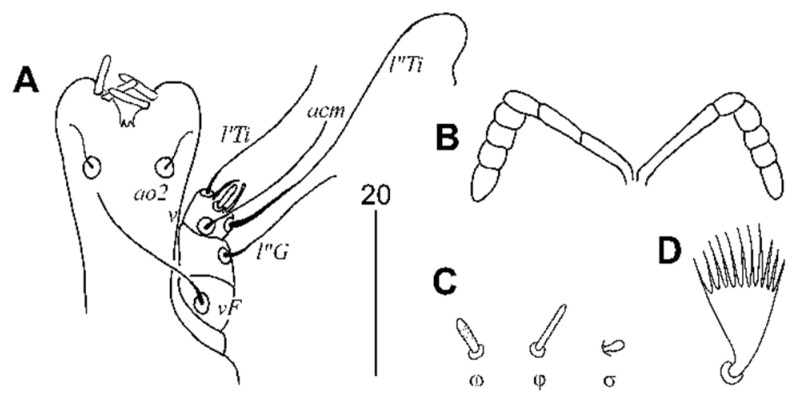
*Peristerophila mayri* Skoracki and Hromada sp. n., female. (**A**)—gnathosoma in ventral view; (**B**)—peritremes; (**C**)—solenidia of leg I; (**D**)—fan-like seta p’III.

**Figure 4 animals-11-03500-f004:**
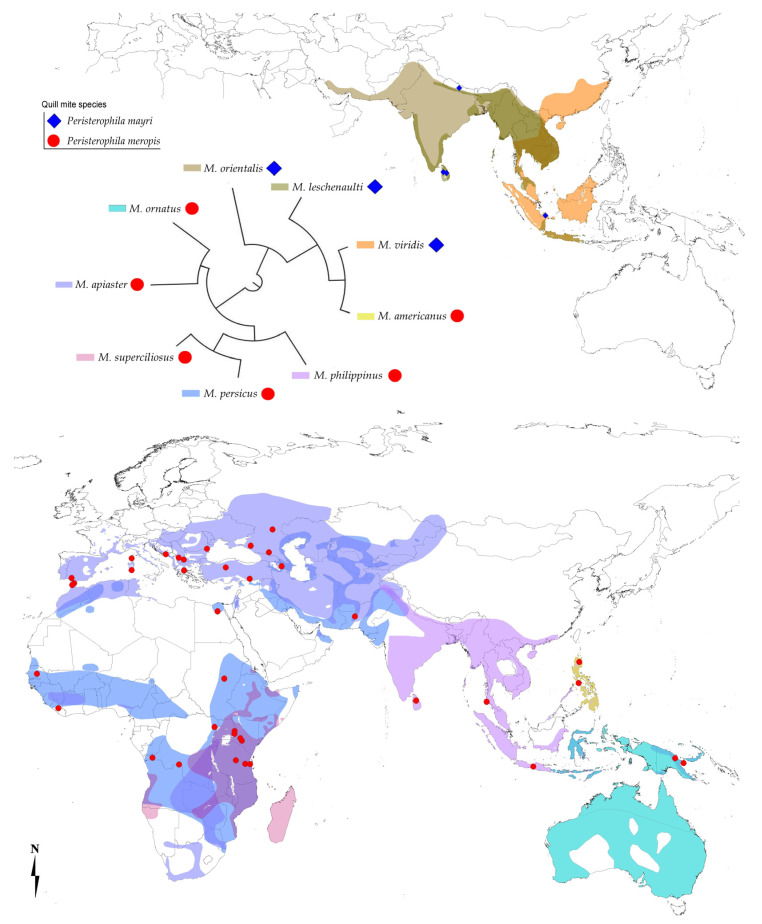
Phylogeny of the *Merops* bee-eater lineage under study, the species’ geographic ranges and recorded localities for the two *Peristerophila* quill mite species (subclades according to Marks et al. [9], adjusted according to Clements et al. [7]; Winkler et al. [8]).

**Table 1 animals-11-03500-t001:** *Peristerophila* quill mites found on examined *Merops* host species and index of prevalence; (*) data from Skoracki et al. [13].

*Peristerophila* Quill Mite Species	*Merops* Hosts	Origin of Host Specimens (Collection)	Host Examined/Infested; Index of Prevalence (IP); Confidence Interval 95% - Sterne (CI)
*P. meropis*	*M. ornatus*	Papua New Guinea (ZSM)	25/4; IP 16.0%; CI 5.7–35.7
*P. meropis*	*M. apiaster* ^(*)^	Spain, France, Gibraltar, Italy, Macedonia, Romania, Bosnia and Herzegovina, Greece, Turkey, Russia, Azerbaijan, Pakistan, Morocco, Tanzania, Kenya (ZSM, NMK)	75/22; IP 29.3%; CI 20–40
*P. meropis*	*M. persicus*	Sudan, Tanzania, Liberia, Senegal, Kenya, D.R. Congo (ZSM, RMCA)	68/10; IP 14.7%; CI 7.8–25.6
*P. meropis*	*M. superciliosus*	Egypt, Tanzania (ZSM, RMCA)	34/13; IP 38.2%; CI 23.2–56.0
*P. meropis*	*M. philippinus*	Thailand, Sri Lanka, Indonesia (ZSM, NMP)	8/3; IP 37.5%; CI 11.1–71.1
*P. meropis*	*M. americanus*	Philippines: Luzon (ZFMK)	8/3; IP 37.5%; CI 11.1–71.1
*P. mayri* sp. n.	*M. viridis*	Philippines: Luzon (ZSM)	10/1; IP 10.0%; CI 0.5–44.7
*P. mayri* sp. n.	*M. leschenaulti*	Nepal, Sri Lanka (ZSM)	13/2; IP 15.4%; CI 2.8–43.4
*P. mayri* sp. n.	*M. orientalis*	Sri Lanka (ZSM)	32/1; IP 3.1%; CI 0.2–16.6

Abbreviations: NMK—National Museum, Nairobi, Kenya; RMCA—Royal Museum for Central Africa, Tervuren, Belgium; NMP—National Museum, Praha, Czech Republic; ZFMK—Zoological Research Museum Alexander Koenig, Bonn, Germany; ZSM—Bavarian State Collection of Zoology, Munich, Germany.

## Data Availability

All necessary data (such as localities) are available in the text of this article.

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
