# Peer review of "The Occurrence of Quill Mites (Arachnida: Acariformes: Syringophilidae) on Bee-Eaters (Aves: Coraciiformes: Meropidae: Merops) of Two Sister Clades"

_animals, 2021, doi:10.3390/ani11123500_

Round 1

Reviewer 1 Report

This is a very nice paper that I enjoyed reading. I cannot criticize this work, it is well designed, clever, and well-written. Congratulations.

The authors investigated the relationship between a clade of host birds (Bee-eaters) and their Syringophilid (quill mite) parasites. They identified (named) the host-parasites species pairs, and this effort included the formal description of a new mite species. Further, they summarized the former and the new information about the geographic distributions of host and parasite species. Finally, they identified a historical host-switch event, as mirrored by the present-day host-parasite phylogenies and geographic distributions. They suggest that this host switch might have been occurred due to the shared breeding colonies of two host species. 

The first author, Maciej Skoracky is probably the most prolific author in the research of quill mites. Up to recently however, he and his co-authors have mostly published taxonomic species descriptions, and our knowledge about the evolution and ecology of this taxon is close to zero. This is, therefore, an important new step to this direction. 

As far as I know, no former authors ever investigated the biogeography of this parasite taxon.

I would recommend carrying out similar studies using other avian and quill mite taxa as well. Increase the number of species involved, not in this paper, but in the future ones. 

The conclusions are well based on factual evidence. There is a hypothesis mentioned in the discussion, but that is clearly labelled as such, it is not at all confused with the facts.

Author Response

Dear reviewer,

thank you very much for your very kind words. Following your suggestions, we even extended our discussion regarding biogeography. We also improved our map and phylogeography, we hope it is even better readable.

Reviewer 2 Report

The manuscript is well written and all the goals are presented very clearly. The species description and figures are very meticulous, and the discussion is very interesting for the reader.  I recommend publication of the manuscript.

The main aim of the studies is investigation of quill mites associated with hosts, i.e. bee-eaters (which represent one clade on phylogenetic tree) and investigation of possible host switches of the mite fauna. 

Other publication focus only on plain species description. Here, the authors for the first time analyze distribution of the mites on closely related hosts (based on their phylogeny) therefore detect possible host switches. Such studies including ecology of the quill mites has never been done before.

For the future studies the authors might include SEM photos which might improve visual part of the manuscript. However, the drawing of the species is very meticulous and clear for the other researcher. The sample size is very big (265 host species) and the methodology is correct.

The hypothesis of possible host-switch is very interesting for the reader and supported by evidence (such as biology/distribution).

The references are appropriate and all the same. There is only 39 cited papers of which 11 papers are citation of one of the authors. But taking into account that there is not so many papers considering this group of parasites written by other authors the number seems reasoned. The authors avoided self-citation by not giving many examples of their previous work.

I would enlarge the figure 4. It should be on the whole page because after printing the host names are barely visible.

Author Response

Dear reviewer,

thank you very much for your kind words. Following your suggestions, we added some, in our opinion, interesting and relevant ideas on the ecology of investigated species.

As we little bit expanded discussion, also the list of references is longer now.

Moreover, we agree that the Figure with maps is hardly readable, therefore, we redo it substantially and asked the journal for enlarging it properly.

Reviewer 3 Report

The paper by Skoracki et al. reports data regarding mites of the family Syringophilidae parasitizing bee-eaters of the genus Merops and the description of a new species within the genus Peristerophila. The paper is original, deals with topics covered by the Journal and is mostly well-written and adequate in English language (some suggested changes are reported below). Methods are adequate too. The paper is particularly interesting as, beyond describing a new species, it deals with the association of parasite and host species and give a new insight in the zoogeography of these parasitic mites.

The only general concern regarding the manuscript is the inconsistent an incorrect use of abbreviations in the genus names of both mites and bird species. The same species is at times (randomly I would say) written with the genus name in full and at times abbreviated. The rule is that the first time a species is mentioned, genus name has to be written in full. After that, it has to be written always abbreviated. Authors must review the whole manuscript, as from this point of view it is plenty of mistakes and very mixed up.

I here down report some specific changes I suggest regarding English language:

  • Line 45: “belong to” should be changed into “are”;
  • Line 46: add a comma after “specificity”;
  • Line 48: add a comma after the closing bracket;
  • Lines 48-49: “throughout the paleotropics…….without Ovceania)”. This sentence is not clear and should be rephrased;
  • Line 61: delete “to” at the beginning of point iii;
  • Line 92: “the investigation the host phylogenies” has to be changed into “the investigation of the host phylogenies”. An “of” has to be added;
  • Line 99: add a comma after “quill mites”;
  • Line 104: change “discovered” into “detected” or “highlighted the presence of”;
  • Line 106: add a comma after Merops;
  • Line 109: here, and in many other parts of the manuscript, why genus name Peristerophila is given in full? Check the whole manuscript on this point;
  • Lines 188-189: “arguably the preeminent evolutionary biologist” maybe “the most preeminent”?;
  • Line 274: add a comma after “oligoxenous species”;
  • Lines 277: add a comma after “of their hosts”;
  • Lines 293-294: this sentence should be rephrased, as it is not correct English.

In conclusion, in my opinion the paper is interesting and deserves publication on Animals, after the minor changes here suggested.

Author Response

Dear reviewer,

thank you very much for your invaluable comments and suggestions. We improved our English according to your suggestions. 

Please find our point-by-point answers:

  • The only general concern regarding the manuscript is the inconsistent an incorrect use of abbreviations in the genus names of both mites and bird species. The same species is at times (randomly I would say) written with the genus name in full and at times abbreviated. The rule is that the first time a species is mentioned, genus name has to be written in full. After that, it has to be written always abbreviated. Authors must review the whole manuscript, as from this point of view it is plenty of mistakes and very mixed up. 
  • THANK YOU, WE IMPROVED THE NAMES
  •  
  • Line 45: “belong to” should be changed into “are”; DONE
  • Line 46: add a comma after “specificity”; DONE
  • Line 48: add a comma after the closing bracket; DONE
  • Lines 48-49: “throughout the paleotropics…….without Ovceania)”. This sentence is not clear and should be rephrased; IMPROVED
  • Line 61: delete “to” at the beginning of point iii; DONE
  • Line 92: “the investigation the host phylogenies” has to be changed into “the investigation of the host phylogenies”. An “of” has to be added; DONE
  • Line 99: add a comma after “quill mites”; DONE
  • Line 104: change “discovered” into “detected” or “highlighted the presence of”; CHANGED
  • Line 106: add a comma after Merops; DONE
  • Line 109: here, and in many other parts of the manuscript, why genus name Peristerophila is given in full? Check the whole manuscript on this point; IMPROVED
  • Lines 188-189: “arguably the preeminent evolutionary biologist” maybe “the most preeminent”?; ADDED
  • Line 274: add a comma after “oligoxenous species”; DONE
  • Lines 277: add a comma after “of their hosts”; DONE
  • Lines 293-294: this sentence should be rephrased, as it is not correct English. REPHRASED